# Adherence to Dietary Recommendations after One Year of Intervention in Breast Cancer Women: The DIANA-5 Trial

**DOI:** 10.3390/nu13092990

**Published:** 2021-08-27

**Authors:** Eleonora Bruno, Vittorio Krogh, Giuliana Gargano, Sara Grioni, Manuela Bellegotti, Elisabetta Venturelli, Salvatore Panico, Maria Santucci de Magistris, Bernardo Bonanni, Emanuela Zagallo, Angelica Mercandino, Maria Chiara Bassi, Rosalba Amodio, Maurizio Zarcone, Rocco Galasso, Maggiorino Barbero, Milena Simeoni, Maria Piera Mano, Franco Berrino, Anna Villarini, Patrizia Pasanisi

**Affiliations:** 1Department of Research, Fondazione IRCCS Istituto Nazionale dei Tumori di Milano, 20133 Milano, Italy; eleonora.bruno@istitutotumori.mi.it (E.B.); vittorio.krogh@istitutotumori.mi.it (V.K.); sara.grioni@istitutotumori.mi.it (S.G.); manuela.bellegotti@istitutotumori.mi.it (M.B.); elisabetta.venturelli@istitutotumori.mi.it (E.V.); francoberrino@gmail.com (F.B.); anna.villarini@istitutotumori.mi.it (A.V.); patrizia.pasanisi@istutotumori.mi.it (P.P.); 2AOU Federico II, 80131 Napoli, Italy; spanico@unina.it (S.P.); masantuc@unina.it (M.S.d.M.); 3Division of Cancer Prevention and Genetics, European Institute of Oncology IRCCS, 20141 Milano, Italy; bernardo.bonanni@ieo.it (B.B.); emanuela.zagallo@ieo.it (E.Z.); 4Fondazione Edo ed Elvo Tempia ONLUS, 13900 Biella, Italy; angelica.mercandino@fondazionetempia.org; 5Ats Valpadana, 46100 Mantova, Italy; mariachiarabassi@gmail.com; 6Clinical Epidemiology and Cancer Registry Unit, Palermo Province Cancer Registry, Palermo University Hospital P. Giaccone, 90127 Palermo, Italy; amodiorosalba@libero.it (R.A.); zarcone7@gmail.com (M.Z.); 7Unit of Regional Cancer Registry, Clinical Epidemiology and Biostatistics, IRCCS-CROB, Basilicata, 85028 Rionero in Vulture, Italy; rocco.galasso@crob.it; 8Obstetrics and Gynecology Unit, Cardinal Massaia Hospital, 14100 Asti, Italy; mbarbero@asl.at.it; 9Associazione LUMEN, 29010 San Pietro in Cerro, Italy; ricerca@naturopatia.org; 10Dipartimento Scienze Chirurgiche, Study University, 10124 Turin, Italy; mariapiera.mano@unito.it; 11S.C. Epidemiologia dei Tumori, AOU Città della Salute e della Scienza, CPO Piemonte, 10126 Turin, Italy

**Keywords:** DIANA-5, adherence to diet, weight and metabolic syndrome improvement

## Abstract

The Diet and Androgen-5 (DIANA-5) trial aimed at testing whether a dietary change based on the Mediterranean diet and on macrobiotic principles can reduce the incidence of breast cancer (BC)-related events. We analyzed the adherence to the DIANA-5 dietary recommendations by randomization group after 1 year of intervention. We evaluated the association between dietary adherence and changes in body weight and metabolic syndrome (MS) parameters. BC women aged 35–70 years were eligible. After the baseline examinations, women were randomized into an intervention group (IG) or a control group (CG). A total of 1344 BC women (689 IG and 655 CG) concluded the first year of dietary intervention. IG showed greater anthropometric and metabolic improvements compared to CG. These changes were significantly associated with increased adherence to the dietary recommendations. Women who increased recommended foods consumption or reduced discouraged foods consumption showed an Odds Ratio (OR) of 1.37 (0.70–2.67) and 2.02 (1.03–3.98) to improve three or more MS parameters. Moreover, women in the higher category of dietary change showed a four times higher OR of reducing body weight compared to the lower category (*p* < 0.001). The DIANA-5 dietary intervention is effective in reducing body weight and MS parameters.

## 1. Introduction

Weight control and physical activity are key points for the management of breast cancer (BC). According to the most recent observational evidence from the World Cancer Research Fund/American Institute for Cancer Research (WCRF/AICR), maintaining healthy body weight, being physically active, following a fiber- and soy-rich diet, and limiting the intake of fat (in particular, saturated fatty acids) may reduce BC co-morbidities (e.g., obesity, hypertension, hyperlipidemia, and diabetes mellitus) and may improve overall survival after BC diagnosis [1].

Previous small trials showed that an insulin-lowering diet significantly decreases body weight, serum testosterone, the bioavailability of both estrogens and IGF-I and the main factors of the metabolic syndrome (MS) in both healthy postmenopausal women and in BC patients [2,3]. These factors are all risk factors for BC and BC recurrences [4,5].

The multi-center randomized controlled trial Diet and Androgen-5 (DIANA-5) [6] aimed at testing the hypothesis that a lifestyle change based on the Mediterranean diet and on macrobiotic principles, together with the recommendation of moderate physical activity, can reduce the incidence of additional BC-related events. Preliminary findings from the DIANA-5 study showed that the presence of MS at baseline, before starting the dietary intervention, is a negative prognostic factor for BC recurrences [7]. Furthermore, a lower baseline prevalence of MS was associated with greater adherence to the WCRF/AICR recommendations, and this is mainly due to meeting the recommendations on physical activity and on the consumption of plant foods [8].

In this paper, we studied adherence to the DIANA-5 dietary recommendations by a randomization group after 1 year of intervention. Furthermore, we evaluated the association between adherence to the DIANA-5 dietary recommendations and changes in body weight and MS parameters.

## 2. Materials and Methods

Detailed information regarding the DIANA-5 study design and protocol has been previously reported [6].

Briefly, women aged 35–70 years, diagnosed with invasive BC within the previous 5 years, free from recurrences or metastases or other cancers, were eligible for the study. All the women were fully informed about the aim of the study, signed an informed consent form, and received general recommendations for the prevention of cancer (the 2007 WCRF/AICR recommendations).

At baseline, the women recruited into the study were requested:To provide copy of the clinical and pathological notes concerning their BC;To complete a questionnaire on their medical history and major cancer risk factors, such as menstrual history and reproductive and behavioral factors (e.g., oral contraceptive use, smoking habits);To fill in 24 h food frequency diary about the previous day’s food intake;To attend a visit for anthropometric and body composition measurements;To provide a 20 mL blood sample for metabolic/hormonal assays (plasma glucose, triglycerides, total, LDL and HDL cholesterol, serum insulin, and serum testosterone);To agree to participate in the dietary intervention in case of randomization to the IG.

After the baseline examinations, the women at high metabolic and hormonal risk of BC recurrences (ER-negative tumors or serum testosterone level ≥ 0.4 ng/mL (1.338 nmol/mL) and/or serum insulin ≥ 7 μU/mL (50 pmol/L) and/or presence of MS) were randomized to an active dietary intervention group (IG) or to a control group (CG) that followed the baseline WCRF/AICR recommendations.

Data collection and measurements were performed in each study center at baseline and at the end of the first, third, and fifth year, and were recorded in a central electronic database at the Fondazione IRCCS Istituto Nazionale dei Tumori of Milan.

The study was approved by the Institutional Review Board and Ethical Committee of the coordinating center Fondazione IRCCS Istituto Nazionale dei Tumori of Milan for all the other collaborating Institutions (INT 37/07). Additionally, the number of the trial is NCT05019989.

### 2.1. Dietary Intervention

After randomization, only the women randomized into the IG were invited to attend dietary activities. The intervention consisted of cooking classes, common meals, and dietary reinforcement meetings once a month during the first year. IG women received the menus supplied during the cookery classes and recipes.

The dietary intervention was based on Mediterranean diet principles and recipes and including some fermented food from Macrobiotic tradition (miso, soy sauce, tempeh, umeboshi) to facilitate digestion and serve as a reliable reservoir for microbiota [9], aimed at:Reducing caloric intake through the preferred consumption of highly satiating foods (whole grain cereals, legumes, and vegetables);Reducing high glycemic index food (refined flours, potatoes, white rice, corn flakes) and high insulinemic foods (sugar and milk). Nuts and legume flours were proposed in sweet and savory cookery recipes. Desserts were prepared without adding sugars, but using instead nuts, fresh fruit, dates, and small amounts of dates, raisins, and dried apricots;Reducing sources of saturated fats (red and processed meat, milk, and dairy products) and avoiding trans fatty acids (margarines and industrial snacks and pastries). Cold-pressed extra-virgin olive oil was the main source of fat. The consumption of nuts and seeds was encouraged;Reducing protein intake, mainly animal proteins. Among animal food, fish, especially cold-water fish (e.g., salmon and mackerel), rich in omega-3 polyunsaturated fatty acids, was privileged.

An example of the proposed menu is shown in Appendix A. Nutritional requirements and most menus and recipes were standardized among the study centers. A nutritionist was always present in the cooking classes and common lunches in order to ensure the necessary assistance to the cooks and to reinforce recommendations to the participants.

### 2.2. Dietary Data Collection

Women’s compliance to the DIANA-5 dietary recommendations was assessed through 24 h food frequency diaries. The participants had to complete the 24 h food frequency diaries at baseline, before starting the dietary intervention, and at the end of the first year (±2 months).

The 24 h food frequency diary contains a list of 65 food items, without any information on portion size or weight or recipes. The women only had to indicate whether, on the previous day, they had eaten or not eaten the specified food at breakfast, lunch, dinner, and breaks. The diary is organized into the following six groups:

**Drinks** (with and without alcohol, with and without added sugar), **milk, and dairy products**. This group includes a specific item for plant-based unsweetened beverages.

**Sweets and confectionery**. This group includes separate items for sugar, whole sugar, honey, and malt.

**Bread and grains**. This group includes separate items for refined grains, whole grains, and grain products.

**Meat, fish, eggs, and meat substitutes**. This group includes separate items for red and processed meat, white meat, and meat substitutes made from wheat gluten protein.

**Legumes, vegetables, fresh and dried fruit, nuts, and seeds.** This group includes separate items for soy products.

**Sauces, animal, and vegetable fats**. This group includes a separate item for miso and soy sauce.

### 2.3. Statistical Analysis

Among 2132 recruited BC women, 590 did not show, after the baseline examinations, a high metabolic/hormonal risk of BC recurrences and did not enter in the randomization. Sociodemographic, clinical, metabolic, and dietary characteristics of 1542 women at high risk were summarized by randomization group using mean and standard deviation (SD), and they were compared by using *t* tests. The frequencies of MS parameters by randomization group were compared by using χ^2^ test. MS was defined on the basis of the presence of at least three components out of five, according to the threshold proposed by the International Diabetic Federation (waist circumference ≥ 80 cm; systolic blood pressure ≥ 130 mmHg, or diastolic blood pressure ≥ 85 mmHg, fasting plasma glucose ≥ 100 mg/100 mL (5.6 mmol/L) or previously diagnosed type II diabetes, triglycerides ≥ 150 mg/100 mL (1.7 mmol/L), high-density lipoprotein < 50 mg/100 mL (1.03 mmol/L), or on the basis of receiving treatment for these components [10].

Body mass index (BMI) was defined as body weight/squared height (kg/m^2^).

Data from the 24 h food frequency diaries about the previous day’s consumption were reported as single food and food groups. On the basis of the proposed DIANA-5 dietary recommendations, we created groups of recommended or discouraged food by summing single food items that equally contribute on the basis of their frequency of consumption.

We created the following groups of recommended foods:Whole-grain products (whole bread, whole rice, other whole grain cereals, unsweetened muesli, oat flakes);Unsweetened beverages (vegetable milk, tea, barley coffee);Vegetables (all vegetables except potatoes);Fruit (all kinds of fruit);Legumes and soy products (legumes, tofu/tempeh);Recommended animal protein products (fish, mollusks, and crustaceans);Dried fruit (apricots, raisins, plums);Nuts and seeds (hazelnuts, almonds, walnuts, nut creams, and oilseeds);Vegetable oils (extra virgin olive oil, seeds oil);Spices and seasonings (miso, tamari, spices);

We created the following groups of discouraged foods:Sugary beverages (sugary beverages, animal milk);Alcoholic drinks (wine, beer, spirits);Sweets and cakes (white sugar, artificial sweeteners, chocolate, candies, biscuits, ice creams, brioches);Refined cereals (white bread, white rice, egg noodles, corn flakes, sweetened muesli);Discouraged animal protein products (red meat, processed meat);Discouraged vegetable protein products (seitan);Dairy products (all kinds of fresh or seasoned cheese, including pizza or pasta);Starchy vegetables (potatoes, mashed potatoes, French fries);Butter and other discouraged seasonings (butter, lard cream, margarine, ready sauces, mayonnaise, ketchup).White meat, eggs, coffee, unsweetened citrus juices, and unsweetened fruit juices were considered neutral food (not recommended but not discouraged).

We created the variable of Recommended Food Consumption/day (RFC) by putting together all the recommended foods (37 items) and the variable of Discouraged Food Consumption/day (DFC) by putting together all the discouraged foods (23 items) as indicators of adherence to the DIANA-5 dietary recommendations.

We also created a Dietary Compliance Indicator (DCI) given by the ratio of RFC and DFC.

A two-sample *t* test was used to compare baseline and 1-year measurements in the two randomized groups.

We analyzed the magnitude of changes in anthropometric, metabolic, and dietary variables by using the difference (delta, Δ) between the 1-year and the baseline measurements for each woman in the two groups.

Changes in MS parameters (ΔMS parameters) were described according to four categories:

No change, worsening of the number of MS parameters, low improvement (reduction of 1 or 2 MS parameters), high improvement (reduction of 3 or more MS parameters).

The association between adherence to the DIANA-5 dietary recommendations expressed as the increase of ΔRFC (tertiles) or the reduction of ΔDFC (tertiles) and the ΔMS parameters was studied by a multinomial logistic regression model and 95% confidence intervals (CI) within the total population at the end of the 1-year dietary intervention. Center, age, BMI at baseline, education (years), hormonal treatment (yes/no), years from BC diagnosis, and chemotherapy (yes/no) in the past were used as model covariates.

Furthermore, a logistic regression model was used to compute the OR and 95%CI of Δbody weight (reduction above the median value, −1.19 kg) by means of the increase of ΔRFC (tertiles) and the reduction of ΔDFC (tertiles) within the total population and by randomization group.

This analysis was repeated by combining the tertiles of ΔRFC and the tertiles of ΔDFC in a unique index of compliance of 9 categories (reference category of low compliance: first tertiles of ΔRFC and first tertiles of ΔDFC). Center, age, weight at baseline, education (years), hormonal treatment (yes/no), years from BC diagnosis, and chemotherapy in the past (yes/no) were used as model covariates.

A *p*-value of <0.05 was taken as significant. All statistical tests were two-sided. Analyses were completed using the STATA 14 statistical package.

## 3. Results

Among the 1542 women at high risk, 769 (mean age 52.0 ± 8.5 SD) were randomized into the IG and 773 (mean age 51.7 ± 8.3 SD) into the CG. On average, the two randomized groups had received a BC diagnosis about 1.8 years before recruitment, with a fairly homogeneous distribution for each year of the 5-year diagnosis–enrolment window (data not shown).

### 3.1. Baseline Metabolic and Anthropometric Characteristics of the Study Population by Randomization Group

Table 1 reports the baseline clinical, metabolic, and anthropometric characteristics of the study population by randomization group. There were no significant differences between the study groups about clinical, sociodemographic, and metabolic variables, but each metabolic parameter was slightly less favorable in the IG so that the prevalence of MS was somewhat higher (40.1% vs. 35.2%, *p* = 0.05).

### 3.2. Baseline Dietary Daily Frequency of Consumption of Recommended and Discouraged Food by Randomization Group

Figure 1 shows the baseline data from the 24 h food frequency diaries. The two groups were homogeneous in regard to the consumption of recommended and discouraged food. At baseline, the IG showed a slightly higher consumption of alcoholic drinks (*p* = 0.22) and recommended animal protein products (*p* = 0.21), while the CG showed a slightly higher consumption of red/processed meat (*p* = 0.29), sugary beverages (*p* = 0.32), and butter and discouraged seasonings (*p* = 0.10). The IG and the CG were also homogenous for RFC/day, DFC/day, and DCI.

After the randomization, 45 women relapsed in the first year; 21 changed their minds, thus dropping out of the study without any dietary intervention; 133 women, for familiar/work reasons, did not participate in the 1-year follow-up examinations.

Therefore, 1344 women (87.1%) participated in the first year clinical follow-up visit (689 in the IG and 655 in the CG).

### 3.3. Changes of Food Frequencies Consumption by Randomization Group

Figure 2 shows the analysis of the Δfood frequencies of consumption. Both groups significantly improved the consumption of almost all recommended foods and reduced the discouraged foods, but the women in the IG showed greater changes.

IG improved the consumption of the recommended foods, i.e whole grain products (*p* < 0.001), unsweetened beverages (*p <* 0.001), legumes (*p* < 0.001), vegetables (*p =* 0.01), dried fruits (*p* < 0.001), nuts and seeds (*p* < 0.001), recommended spices (*p* < 0.001), and in general improved the RFC more than the CG. Consistently, the IG reduced sugary beverages (*p* < 0.001), alcoholic drinks (*p* < 0.001), sweets and cakes (*p* < 0.001), refined cereals (*p* < 0.001), and red and processed meat (*p* = 0.01).

Overall, the IG improved the RFC (+3.8 vs. +1.3; *p* < 0.001) more than the CG and significantly reduced DFC (−3.0 vs. −1.8; *p* < 0.001) compared to the CG (Table 2). Consistently, the IG showed a strong significant improvement of DCI compared to the CG.

### 3.4. Changes of Anthropometric and Metabolic Parameters by Randomization Group

Regarding the anthropometric and metabolic parameters, both groups showed a significant improvement, but the IG showed changes with greater impact for health in all the parameters under study (Table 2). The IG lost weight (*p* < 0.001), BMI (*p* < 0.001), and lowered waist circumference (*p* < 0.001), glycemia (*p* = 0.04), and triglycerides (*p =* 0.01) significantly more than the CG. Consistently, the reduction of the MS parameters was significantly stronger in the IG compared to the CG (*p* < 0.001).

### 3.5. Changes in ΔMS Parameters According to the Δ of Recommended Food Consumption and Discouraged Food Consumption after One Year of Dietary Intervention

Regarding the effect of the adherence to the DIANA-5 dietary recommendations, Table 3 report the ORs (95% CIs) of change in MS parameters according to the increase of ΔRFC (3a) and the reduction of ΔDFC (3b) after 1 year of dietary intervention.

The women who increased more than five frequencies of recommended foods consumption (higher tertile of ΔRFC) (3a) showed an OR of 1.37 (0.70–2.67) to improve three or more MS parameters than women who did not change MS parameters. Conversely, the women in the higher tertile of ΔRFC showed a lower OR of worsening MS parameters (OR: 0.75; 0.39–1.43).

The women who reduced more than four frequencies of discouraged food consumption (higher tertile of ΔDFC) (3b) showed an OR two times higher of improving three or more MS parameters than the women who did not change (*p* = 0.04).

### 3.6. Association between the Adherence to DIANA-5 Dietary Recommendations and Δbody Weight

Regarding the association between the adherence to DIANA-5 dietary recommendations and Δbody weight (reduction above the median value), all women significantly increased the ORs of losing their body weight by increasing the ΔRFC (OR: 1.99; 1.47–2.70, *p* < 0.001), and by reducing ΔDFC (OR = 2.79; 2.05–3.80, *p* < 0.001). Consistently, women in the higher tertile of ΔRFC and ΔDFC experienced the higher reduction in body weight (−2.8 kg versus −0.6 kg) compared to the women in the lower category of dietary change. The IG showed greater changes in body weight compared to CG (−3.9 kg versus −2.4 kg, *p* < 0.001).

Table 4 reports the ORs (95% CIs) of the Δbody weight (reduction above the median value) by combining the tertiles of ΔRFC and the tertiles of ΔDFC.

Women in the higher category of dietary change (third tertile of ΔRFC and third tertile of ΔDFC) showed a four times higher OR of reducing body weight compared to women in the lower category (*p* < 0.001).

We also observed a significant improvement of body weight in women in the intermediate categories of adherence (second tertile of ΔRFC and third tertile of ΔDFC (−2.5 kg); second tertile of ΔRFC and second tertile of ΔDFC (−2.4 kg)).

Furthermore, analyzing the extreme categories of dietary change, we observed that women who strongly reduced the consumption of discouraged food but did not increase the consumption of recommended food (first tertile of ΔRFC and third tertile of ΔDFC) experienced an OR of 1.18 of reducing their body weight. Conversely, the women who strongly increased the consumption of ΔRFC but did not reduce their consumption of discouraged food (third tertile of ΔRFC and first tertile of ΔDFC) showed a lower OR of reducing their body weight above the median value (OR:0.74; 0.39–1.40, *p* = 0.36).

## 4. Discussion

The findings from this large Italian randomized controlled trial suggest that a dietary intervention based on the Mediterranean diet and on macrobiotic dietary principles is effective in reducing body weight and MS parameters in BC women at high metabolic/hormonal risk of BC recurrences. The women randomized in the IG showed greater improvements in their anthropometric and metabolic characteristics compared to CG. These changes were significantly associated with increased adherence to the DIANA-5 dietary recommendations.

Although observational evidence suggested that some foods (red and processed meat) and nutrients (saturated fatty acids) are potential risk factors for BC, given their effect on insulin resistance and inflammation markers, obesity and alcohol consumption are the only factors that were consistently found to be associated with BC incidence, and obesity was also found to be associated with BC recurrences and BC mortality. The two largest trials about BC women, the Women’s Intervention Nutrition Study (WINS) and the Women’s Healthy Eating and Living (WHEL), designed their dietary intervention mainly to lower participants’ fat intake from 30% to 20% of total calories [11,12]. Additionally, in the WHEL trial, participants were requested to increase their vegetable and fruits servings in order to improve fiber intake. However, body weight only significantly decreased in the WINS study, and subsequently, the authors observed a reduction in BC recurrences.

The DIANA-5 dietary intervention consisted of a comprehensive qualitative dietary change, with a moderate calorie restriction obtained by encouraging the consumption of highly satiating food (unrefined cereals, legumes, and vegetables) and by avoiding the consumption of calorie-dense processed food, sweets, and sugary drinks. Furthermore, the women were invited to limit the consumption of red and processed meat (high in saturated fats) and to prefer extra-virgin olive oil, nuts, and oleaginous seeds as a source of unrefined vegetable fats. In our study, the women belonging to the IG significantly increased their adherence to the proposed DIANA-5 dietary recommendations and lost 2.5 kg on average and 2.6 cm in their waist, a difference of the same order as the one obtained in the WINS study [11]. In detail, the IG women with the higher increase of the recommended food consumption (ΔRFC) and the higher reduction of the discouraged food consumption (ΔDFC) obtained the greater improvement in their body weight.

The women in the IG also showed, after 1 year of dietary intervention, an improvement in their metabolic parameters and a reduction of the number of MS factors, particularly in the women who increased the ΔRFC and reduced ΔDFC.

Furthermore, our results found a major impact of ΔDFC on MS parameters, which suggests that increasing recommended foods is not sufficient to improve metabolic markers if discouraged foods are not decreased as well.

MS increases BC relapses [5,7]. The mechanisms by which MS affects prognosis are likely to include higher sex hormone levels, higher levels of insulin and IGF-I, and chronic inflammatory status [5,13,14,15,16]. The women in the IG obtained an MS regression of about 25%. Compared to other trials of the Mediterranean diet and MS [17,18,19,20], we observed a lower MS regression after 1 year of dietary intervention. However, in the present study, we did not work with “healthy” dysmetabolic people. BC women suffer from special issues and problems, such as treatment-related adverse effects, fatigue, and depression, which can complicate weight management and the control of metabolic and hormonal parameters. In detail, premature menopause is a consequence of chemotherapy and/or hormonal treatment, and it is accompanied by physiologic changes that can affect energy balance, with a consequent increase in abdominal fat, reduction in lean mass, and worsening of MS parameters [21]. In our study, we did not find any significant interaction with hormonal treatments. Overall, our dietary intervention seemed to be more effective in the reduction of body weight, waist circumference, and triglycerides, while it obtained only minor changes in blood pressure, glycemia, and HDL cholesterol. Higher adherence to the DIANA-5 recommendations was helpful for the management of weight and for the control of MS parameters in BC women. Interestingly, the results obtained by combining the two indices of dietary adherence also suggested a protective effect for the intermediate categories of dietary change. In particular, the reduction of the DFC seemed to be more important for the control of body weight and MS parameters than for the increase of the recommended ones.

The major strength of the present study is the large sample size. Furthermore, the DIANA-5 is the only randomized controlled trial that will likely be able to test whether a Mediterranean and macrobiotic dietary intervention may improve BC survival.

One limitation is that women from the CG showed some improvements in the metabolic and anthropometric parameters of interest. These changes were partially expected by design. In fact, all women, on the day they were recruited, attended a conference in which the 2007 WCRF/AICR lifestyle recommendations for cancer prevention were explained and received a standard breakfast that respects WCRF principles [1]. These recommendations and the specific interest of BC women in the benefits of a healthy diet led the CG to make some adjustments in their dietary habits.

A further limitation is that, although the IG received notes on the nutritional content of the menus and recipes, we did not use any dietary instruments to estimate the women’s nutrient intake. However, the 24 h food frequency diaries are extremely efficient to describe and compare the average consumption of specific foods and dietary patterns between populations, in line with our aim [22].

## 5. Conclusions

These results after the 1-year dietary intervention are encouraging. The follow-up of the cohort and the subsequent survival analysis will allow a better understanding of whether the obtained weight reduction and metabolic changes may affect the progression of the disease.

## Figures and Tables

**Figure 1 nutrients-13-02990-f001:**
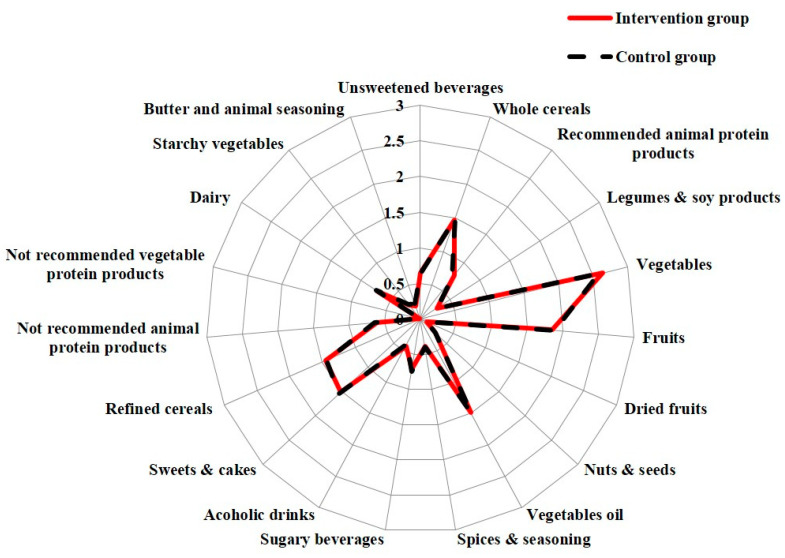
Baseline daily frequency of consumption of recommended and discouraged food by randomization group. The baseline distribution of frequencies of food consumption in the intervention group and in the control group is represented by a Kiviat diagram. This graphic representation consists of a sequence of rays that originate from a center and forms equal angles to each other; each ray represents one of the food/food group variables. The distance from the central point marked on the radius is the maximum achieved frequency of consumption (time/day). The points on the rays are joined with two segments which are red and continuous for the intervention group and dashed for the control group.

**Figure 2 nutrients-13-02990-f002:**
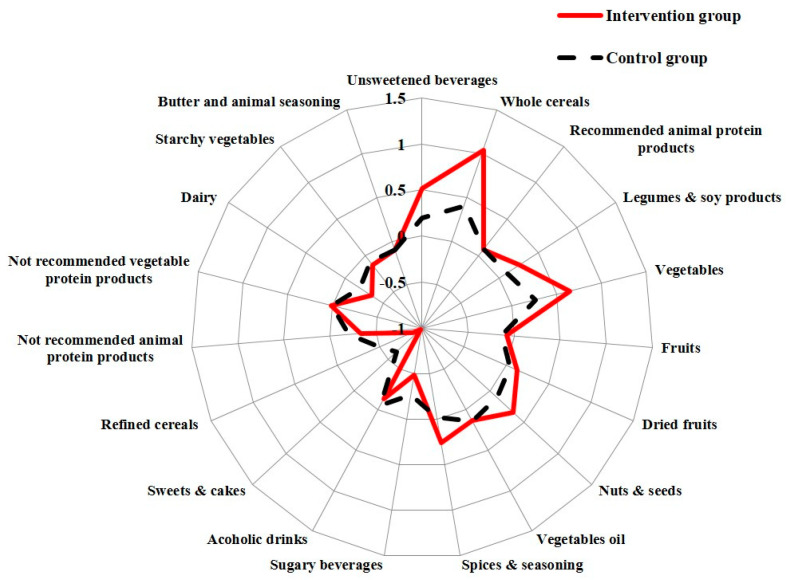
Changes of food frequencies consumption by randomization group. The distribution of the “Delta, Δ” change in frequencies of food consumption (time/day) in the intervention group and in the control group is represented by a Kiviat diagram. This graphic representation consists of a sequence of rays that originate from a center and forms equal angles to each other; each ray represents one of the food/food group variables. The distance from the center of the point marked on the radius is the “Delta, Δ” change of frequencies of food/food group consumption (time/day). The points on the rays are joined with two segments, red and continuous for the intervention group and dashed for the control group.

**Table 1 nutrients-13-02990-t001:** Baseline metabolic and anthropometric characteristics of the study population by randomization group.

	IG	CG	*p **
	(N = 769)	(N = 773)	
**Age at recruitment (years)**	52.0 ± 8.5	51.7 ± 8.3	0.37
**Education (%)**			
**First level**	24.7	25.4	
**Second level**	46	47.8	0.34
**Third level**	29.3	26.8	
**ER-negative (%)**	21.3	20.6	0.62
**Menopause (%)**	40.9	39.4	0.61
**Natural menopause (%)**	80.2	80.2	0.98
**Tumor stage (%)**			
**IA**	43.1	41.8	
**IB**	17.6	17.2	
**IIA**	13.6	14.2	
**IIB**	9.7	11.5	
**IIIA**	9	9.5	
**IIIB**	0.6	1.1	
**IIIC**	6.4	4.7	0.46
**Height (cm)**	160.8 ± 6.3	160.6 ± 6.3	0.52
**Weight (kg)**	68.9 ± 13.0	68.3 ± 12.8	0.28
**BMI (kg/m^2^)**	26.7 ± 5.0	26.5 ± 5.1	0.5
**Waist circumference (cm)**	87.5 ± 12.4	86.8 ± 12.3	0.29
**Systolic blood pressure (mm/Hg)**	127.5 ± 18.5	126.4 ± 18.6	0.26
**Diastolic blood pressure (mm/Hg)**	82.4 ± 11.4	81.6 ± 11.9	0.18
**Fasting glucose (mg/dL)**	94.3 ± 18.1	94.0 ± 16.1	0.67
**HDL (mg/dL)**	60.3 ± 15.6	61.5 ± 15.5	0.13
**Triglycerides (mg/dL)**	113.8 ± 68.0	109.4 ± 61.3	0.18
**Number of MS parameters**			
**0**	12.4	16.2	
**1**	23	22	
**2**	24.6	26.7	
**3**	23	21.4	
**4**	12.3	10.8	
**5**	4.7	2.9	0.1

* *p* of differences using Student’s *t* test for continuous variables and chi-squared test for percentages.

**Table 2 nutrients-13-02990-t002:** Changes of anthropometric and metabolic parameters by randomization group.

	IG			CG			IG	CG	
	Baseline Mean ± SD	1 Year Mean ± SD	*p **	Baseline Mean ± SD	1 Year Mean ± SD	*p **	Δ	Δ	*p ***
	(N = 689)	(N = 689)		(N = 655)	(N = 655)				
**Weight (kg)**	68.9 ± 12.9	66.5 ± 13.1	<0.001	67.7 ± 12.4	66.8 ± 12.5	<0.001	−2.4	−0.9	<0.001
**BMI (kg/m^2^)**	26.6 ± 5.0	25.7 ± 5.0	<0.001	26.2 ± 4.9	25.8 ± 4.9	<0.001	−1.0	−0.4	<0.001
**Waist circ. (cm)**	87.3 ± 12.3	84.7 ± 12.3	<0.001	86.2 ± 12.1	85.2 ± 11.8	<0.001	−2.6	−1.0	<0.001
**Diastolic pressure (mm/Hg)**	82.4 ± 11.4	80.8 ± 11.7	<0.001	81.2 ± 11.9	80.4 ± 10.4	0.02	−1.6	−0.8	0.16
**Systolic pressure (mm/Hg)**	127.7 ± 18.2	125.5 ± 19.1	<0.001	126.0 ± 18.2	125.2 ± 17.8	0.24	−2.2	−0.8	0.1
**Fasting glucose (mg/dL)**	94.2 ± 17.2	92.5 ± 16.3	<0.001	93.7 ± 14.0	93.1 ± 13.7	0.21	−1.7	−0.5	0.04
**HDL (mg/dL)**	60.8 ± 15.7	61.2 ± 16.1	0.28	62.3 ± 15.5	62.7 ± 16.0	0.34	0.4	0.4	0.95
**Triglycerides (mg/dL)**	114.3 ± 70.8	100.3 ± 61.6	<0.001	107.4 ± 59.6	99.2 ± 51.6	<0.001	−14.0	−8.2	0.03
**Number of MS parameters**									
**0**	12.3	18		16.2	18.9		5.7	2.7	
**1**	23	22		22	23.5		−1.0	1.5	
**2**	24.7	29.2		26.7	28		4.5	1.3	
**3**	23	18.6		21.4	16.8		−4.4	−4.6	
**4**	12.3	10		10.8	9.7		−2.3	−1.1	
**5**	4.7	2.2	<0.001	2.9	3.1	<0.001	−2.5	0.2	0.01
**RFC/day ^§^**	10.1 ± 5.4	13.9 ± 5.9	<0.001	10.1 ± 5.3	11.4 ± 5.4	<0.001	1.8	1.3	<0.001
**DFC/day ^§^**	5.6 ± 3.1	2.6 ± 2.8	<0.001	5.6 ± 3.9	3.9 ± 2.9	<0.001	−3.0	−1.8	<0.001
**DCI ^§^**	2.6 ± 2.7	6.9 ± 6.0	<0.001	2.6 ± 2.2	4.3 ± 4.5	<0.001	4.5	1.9	<0.001

RFC = Recommended Food Consumption; DFC = Discouraged Food Consumption; DCI = Dietary Compliance Index. * *p* of before–after analysis using Student’s *t* test for continuous variables and chi-squared test for percentages ** *p* of Δ differences using Student’s *t* test. **^§^** Data were available for 1298 women (659 IG and 639 CG) due to missing values in same food items in the diaries.

**Table 3 nutrients-13-02990-t003:** Adjusted ORs* (95% CIs) of changes in ΔMS parameters according to the Δ of Recommended Food Consumption (a) and Discouraged Food Consumption (b) after 1 year of dietary intervention.

ΔMS Parameters (a)				
**ΔRFC/Day ^§^**	**Worsening**	**No Change**	**Low Improvement**	**High Improvement**
**1st Tertile (<+1)** **(n = 435)**	1	1	1	1
**2nd Tertile (1 to 5)** **(n = 494)**	0.84 (0.46–1.54)	1	0.94 (0.51–1.74)	1.26 (0.67–2.38)
**3rd Tertile (>5)** **(n = 369)**	0.75 (0.39–1.43)	1	0.96 (0.50–1.84)	1.37 (0.70–2.67)
**ΔMS Parameters (b)**				
**ΔDFC/Day ^§^**	**Worsening**	**No Change**	**Low Improvement**	**High Improvement**
**1st Tertile (<−1)** **(n = 356)**	1	1	1	**1**
**2nd Tertile (−3 to –1)** **(n = 478)**	1.02 (0.55–1.88)	1	1.07 (0.58–2.02)	**1.26** **(0.66–2.40)**
**3rd Tertile(>−4)** **(n = 464)**	1.18 (0.61–2.27)	1	1.53 (0.79–2.96)	**2.02** **(1.03–3.98)**

ΔRFC = delta value of Recommended Food Consumption expressed as daily frequencies of consumption; ΔDFC = delta value of Discouraged Food Consumption expressed as daily frequencies of consumption; ΔMS parameters = delta value of Metabolic Syndrome parameters; low improvement = reduction of 1 or 2 MS parameters; high improvement = reduction of 3 or more MS parameters. *ORs adjusted for center, age, BMI at baseline, education (years), hormonal treatment (yes/n o), years from BC diagnosis, and chemotherapy (yes/no) in the past. **^§^** Data were available for 1298 women (659 IG and 639 CG) due to missing values in same food items in the diaries.

**Table 4 nutrients-13-02990-t004:** Adjusted ORs* (95% CI) of Δbody weight (reduction above the median value) according to the tertiles of ΔRFC and ΔDFC after 1 year of dietary intervention.

	1st Tertile ΔRFC (<+1) (n = 435)	2nd Tertile ΔRFC (+1 to +5) (n = 494)	3rd Tertile ΔRFC (>+5) (n = 369)
**1st Tertile** **ΔDFC (<−1)** **(n = 356)**	1	0.67 (0.39–1.13)	0.74 (0.39–1.40)
**2nd Tertile** **ΔDFC (−3 to −1)** **(n = 478)**	1.08 (0.66–1.77)	1.51 (0.94–2.40)	**1.64** **(0.98–2.72)**
**3rd Tertile** **ΔDFC (>−4)** **(n = 464)**	1.18 (0.70–1.97)	**2.10** **(1.29–3.40)**	**4.14** **(2.51–6.28)**

ΔRFC = delta value of Recommended Food Consumption expressed as daily frequencies of consumption; ΔDFC = delta value of Discouraged Food Consumption expressed as daily frequencies of consumption. *ORs adjusted for center, age, weight at baseline (kg), education (years), hormonal treatment (yes/no), years from BC diagnosis, and chemotherapy (yes/no) in the past.

## Data Availability

The data presented in this study are available on request from the corresponding author. The data are not publicly available due to privacy and ethical restiction.

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
