# Peer review of "Adherence to Dietary Recommendations after One Year of Intervention in Breast Cancer Women: The DIANA-5 Trial"

_nutrients, 2021, doi:10.3390/nu13092990_

Round 1
Reviewer 1 Report
Bruno and colleagues studied in the present manuscript the adherence to the DIANA-5 dietary recommendations by randomization group after a one year of intervention. Furthermore, the study group evaluated the assocation between adherence to the DIANA-5 dietary recommendations and changes in body weight and MS parameters.
The present study is a well written and structured manuscript. It is easy to follow and understand. Materials and methods are described in detail and in the introduction section the background of the trial is described in detail.
Figure 1 and figure 2 illustrate the results of the trial very well.
Some little errors can be corrected in the tables:
Table 1
please add a column with p-values for group comparison
Table 2
please delete '.' after diastolic pressure and add blood - 'diastolic blood pressure' - and 'systolic blood pressure' (please change also in table 1)
Typing error: 'tryglicerides'
Please use 'blood glucose' or 'fasting glucose' or 'plasma glucose' instead of 'glycemia' (please also change in table 1)
Typing error: RFC - 'Cunsumption'
In the final discussion section the outcomes of the trial are discussed in detail and the limitations and strength of the trial are also cited.
Thank you for the opportunity to review this interesting manuscript.
Author Response
Dear editor,
Here are reported our answers to reviewer comments.
Table 1 please add a column with p-values for group comparison
- Thank you, we added the column with p-value.
Table 2: please delete '.' after diastolic pressure and add blood - 'diastolic blood pressure' - and 'systolic blood pressure' (please change also in table 1)
- Thank you, we changed it.
Typing error: 'tryglicerides'
- Thank you we corrected it.
Please use 'blood glucose' or 'fasting glucose' or 'plasma glucose' instead of 'glycemia' (please also change in table 1)
- Thank you, we corrected it.
Typing error: RFC - 'Cunsumption'
- Thank you, we corrected it.
Reviewer 2 Report
The current study presented encouraging findings that DIANA-5 dietary intervention reduced body weight and metabolic syndrome (MS) parameters among breast cancer survivors after 1 year of intervention. The manuscript is generally well written. I have some comments below.
- Methods: A bit more details are needed about how recommended food consumption (RFC) and discouraged food consumption (DFC) variables were created – did each food item contribute equally to the calculation? RFI and DFI were also used in the manuscript – do they mean the same thing as RFC and DFC?
- Providing additional sociodemographic info (e.g., education level), menopausal status, and BrCa clinic factors (e.g, stage distribution, ER+/ER-) would help readers better understand the study population.
- Is there any difference in participants’ characteristics between the IG analytical group (n=689) vs. CG analytical group (n=655)?
- In the multinomial logistic regression evaluating change of RFI or DFI with change in MS as well as the logistic regression for weight change, the rationale of covariates choice in the model needs to be provided. What about smoking, physical activity level, breast cancer stage, and ER status – would they influence the associations?
- The authors could consider adding n of each group into table 3 and table 4 to make them more informational.
- The resolution of Figure 1 is a bit low, hard to see the intervention group in the figure. “The distance from the central point …is proportional to …the maximum achievable frequency of consumption”: this interpretation is not easy to follow. In addition, what is the unit of the numbers in the figure (e.g., times/day)?
Author Response
Dear editor,
Here are reported our answers to reviewers’ comments.
- Methods: A bit more details are needed about how recommended food consumption (RFC) and discouraged food consumption (DFC) variables were created – did each food item contribute equally to the calculation? RFI and DFI were also used in the manuscript – do they mean the same thing as RFC and DFC?
Thank you for the comment. Yes, each variable contributes in the RFC and DFC equally. The revised manuscript now includes a sentence in the paragraph 2.3 (Statistical analysis, pag.4) that clarifies this point to readers. Each food item contributes equally with its specific frequency of consumption (frequencies of consumption range 0-4).
About RFI and DFI, we apologize for the typos and we corrected them into the manuscript.
2. Providing additional sociodemographic info (e.g., education level), menopausal status, and BrCa clinic factors (e.g, stage distribution, ER+/ER-) would help readers better understand the study population.
Thank you, we added the sociodemographic and clinical information in Tab.1 (pag. 6)
3. Is there any difference in participants’ characteristics between the IG analytical group (n=689) vs. CG analytical group (n=655)?
No, there are no differences at baseline between the 689 women in the IG and the 655 in the CG.
4. In the multinomial logistic regression evaluating change of RFI or DFI with change in MS as well as the logistic regression for weight change, the rationale of covariates choice in the model needs to be provided. What about smoking, physical activity level, breast cancer stage, and ER status – would they influence the associations?
Many thanks’ for the comment, we used hormonal treatments and previous chemotherapy treatments instead of ER status and breast cancer stage because treatments may influence dietary habits, weight gain and MS parameters.
Following your suggestion, we added into the model smoking status and physical activity levels but the results in the multinomial and logistic regression did not change.
5. The authors could consider adding n of each group into table 3 and table 4 to make them more informational.
Thank you, we added the n of each group in table 2 (pag. XX), table 3 (pag.10) and table 4 (pag.11)
6. The resolution of Figure 1 is a bit low, hard to see the intervention group in the figure. “The distance from the central point …is proportional to …the maximum achievable frequency of consumption”: this interpretation is not easy to follow. In addition, what is the unit of the numbers in the figure (e.g., times/day)?
Thank you for the comment, we changed the color line of the intervention group in figure 1 and 2 to better visualize the two groups.
Furthermore, we simplify the sentence “The distance from the central point …is proportional to …the maximum achievable frequency of consumption” below the figure 1 (pag.7).
The number in the figure are times/day. We added the unit below the figures.